# One-Step Suicide Substrate Inactivation Kinetics of a Ping-Pong Reaction with One Substrate Undergoing Disproportionation: A Theoretical Approach with Approximate Solutions

Ismael Gutiérrez-Fernández [†] , Ouardia Bendou [†], Nara Bueno-Ramos , Emilio L. Marcos-Barbero , Rosa Morcuende and Juan B. Arellano *

Institute of Natural Resources and Agrobiology of Salamanca (IRNASA), Consejo Superior de Investigaciones Científicas (CSIC), 37008 Salamanca, Spain
* Correspondence: juan.arellano@irnasa.csic.es; Tel.: +34-923-386-369
† These authors contributed equally to this work.

**Abstract:** Understanding the kinetic mechanism of enzyme inactivation by suicide substrate is of relevance for the optimal design of new drugs with pharmacological and therapeutic applications. Suicide substrate inactivation usually occurs via a two-step mechanism, although there are enzymes such as peroxidase and catalase in which the suicide inactivation by $H_2O_2$ happens in a single step. The approximate solution of the ordinary differential equation (ODE) system of the one step suicide substrate inactivation kinetics for a uni–uni reaction following the irreversible Michaelis–Menten model was previously analytically solved when accumulation of the substrate–enzyme complex was negligible, however not for more complex models, such as a ping-pong reaction, in which the enzyme is present in two active states during the catalytic turnover. To solve this issue, a theoretical approach was followed, in which the standard quasi-steady state and reactant stationary approximations were invoked. These approximations allowed for solving the ODE system of a ping-pong reaction with one substrate undergoing disproportionation when suicide inactivation was also present. Although the approximate analytical solutions were rather unwieldy, they were still valuable in qualitative analyses to explore the time course of the reaction products and identify the enzyme active state that irreversibly reacted with the suicide substrate during the reaction.

**Keywords:** catalase; enzymatic kinetics; disproportionation reaction; Michaelis–Menten model; ping-pong reaction; reactant stationary assumption; quasi-steady-state approximation; suicide substrate inactivation

**MSC:** 34A05; 92B05; 92C45; 97B10

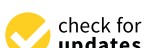

## 1. Introduction

The kinetic analysis of the inhibition of enzyme reactions by inhibitors and inactivators is widely used in enzymology to better understand the basis of the catalytic mechanism via which substrates bind to enzymes and are converted into products [1,2]. Furthermore, it is a frontline strategy to design highly effective enzyme-targeted drugs with pharmacological and therapeutic applications [3–5]. The inhibitors can reversibly compete with the substrates for the binding at the active site, bind to other enzyme sites different from the active site, or alternatively bind to the enzyme when the substrate–enzyme complex is already present [6–8]. Additionally, reversible enzyme inhibition can be complete or partial if, in the latter case, there is still evidence for a nonzero enzyme rate at saturating inhibitor concentrations [7,9].

When irreversible enzyme inhibition occurs, the inactivators can either directly react with the enzyme in a one-step mechanism, or instead, initially reversibly bind to the

enzyme and then react with it in a two-step mechanism, producing in both cases a catalytic dysfunction that lasts for a prolonged period. In most cases, irreversible molecular modification is the formation of a covalent bond between the inactivator and the enzyme, although noncovalent-bonded inhibitors can also be included in this classification if the molecular binding is very tight [10]. Irreversible inhibition includes suicide substrate (or mechanism-based) inactivation, which is characterized by the fact that the substrate of the enzyme also plays the role of the inactivator [11,12]. This communication is focused on the numerical and analytical kinetic analysis of the inactivation of enzymes when the irreversible reaction with the suicide substrate occurs in one single step.

Suicide substrates have received a great deal of attention in drug design because they are unreactive in the reaction medium before binding to the target enzyme, but in contrast, are highly reactive when bound to it. The irreversible modification of the active site by the suicide substrate blocks the catalytic turnover of the enzyme; therefore, product formation and enzyme inactivation become two competing reactions [13,14]. In the absence of enzyme inactivation (or inhibition), the kinetics scheme for an irreversible uni–uni enzyme reaction, in which one single intermediate substrate–enzyme complex EA is formed, follows the well-established Michaelis–Menten model [15,16]:

$$A + E \underset{k_b}{\overset{k_a}{\rightleftarrows}} EA \overset{k_c}{\rightarrow} E + P. \tag{1}$$

The definitions of symbols for the reaction compounds, rate constants, and subscripts of the above enzyme reaction and others below are given in Table 1. The exact integrated solution of the nonlinear ordinary differential equation (ODE) system of this irreversible uni–uni Michaelis–Menten model cannot be derived using analytical methods, and only approximate solutions can be given after applying the standard and total quasi-steady-state approximations [17–19]. Indeed, the integrated solutions for the substrate, intermediate substrate–enzyme complex, and product in the transient and steady-state phases have been a matter of intensive research in the past century [17,20–23], which persists today [18,24].

When there is suicide substrate inactivation, the aforementioned Michaelis–Menten model changes to accommodate the binding and reaction of the inactivator with the enzyme active site. The reaction scheme of the enzyme inactivation model following a two-step mechanism is generally schematized as follows [12]:

$$A + E \underset{k_b}{\overset{k_a}{\rightleftarrows}} EA \overset{k_x}{\rightarrow} EX \overset{k_p}{\rightarrow} E + P, \tag{2}$$

$$EX \overset{k_i'}{\rightarrow} I', \tag{3}$$

where EA is irreversibly converted to EX. This latter intermediate complex can either follow the catalytic turnover and dissociate into the reaction product and the active form of the enzyme or irreversibly become an inactive form of the enzyme.

The kinetic analysis of the above suicide substrate inactivation model was first developed by Waley [13,25]. In the seminal model, the formation of the inactive state of the enzyme (i.e., I') was a first-order reaction, and EA and EX were assumed to be in a steady state. Subsequent studies were conducted in which different conditions were examined within the uni–uni Michaelis–Menten model, including the ratio between the initial concentrations of the substrate (or inactivator) and the enzyme, the presence of auxiliary substrates, the intermediate complex reacting with the inactivator, or the effect of the reaction product [26–31].

**Table 1.** Nomenclature and symbols for the description of the uni–uni and ping-pong reactions.

| Nomenclature | Definition |
|---|---|
| Compounds | |
| A | Reaction (suicide) substrate |
| E | Active enzyme |
| EA | Intermediate substrate–enzyme complex |
| EX | Second intermediate substrate–enzyme complex in the Waley model |
| F | Intermediate active enzyme of a ping-pong reaction |
| FA | Intermediate substrate–enzyme complex of a ping-pong reaction |
| I′ | Inactive enzyme of the Waley model |
| I | Inactive enzyme of a one-step suicide substrate inactivation reaction |
| P | (First) reaction product (of a ping-pong reaction) |
| Q | Second reaction product of a ping-pong reaction |
| Constants | |
| $K_M$ | Michaelis constant of a uni–uni reaction : $(k_b + k_c)/k_a$, mol·m$^{-3}$ |
| $K_M^E$ | Michaelis constant of the first stage of a ping–pong reaction in which A binds to and reacts with E : $k_f(k_b + k_c)/[k_a(k_c + k_f)]$, mol·m$^{-3}$ |
| $K_M^F$ | Michaelis constant of the second stage of a ping–pong reaction in which A binds to and reacts with F : $k_c(k_e + k_f)/[k_d(k_c + k_f)]$, mol·m$^{-3}$ |
| $k_a$ | Second-order forward rate constant, m$^3$·(mol·s)$^{-1}$ |
| $k_{a,\,cat}$ | Second-order rate constant for $H_2O_2$ decomposition, m$^3$·(mol·s)$^{-1}$ |
| $k_b$ | First-order backward rate constant, s$^{-1}$ |
| $k_c$ | First-order forward rate constant, s$^{-1}$ |
| $k_d$ | Second-order forward rate constant, m$^3$·(mol·s)$^{-1}$ |
| $k_e$ | First-order backward rate constant, s$^{-1}$ |
| $k_f$ | First-order forward rate constant, s$^{-1}$ |
| $k'_i$ | First–order forward rate constant for the formation of I′, s$^{-1}$ |
| $k_i$ | Second-order forward rate constant for the formation of I, m$^3$·(mol·s)$^{-1}$ |
| $k_{i,\,cat}$ | Second-order rate constant for catalase inactivation, m$^3$·(mol·s)$^{-1}$ |
| $k_p$ | First-order forward rate constant for the decomposition of EX, s$^{-1}$ |
| $k_x$ | First-order forward rate constant for the formation of EX, s$^{-1}$ |
| $k_E$ | A function of second–order forward rate constants when E is inactivated in a ping–pong reaction : $\left\lvert \sqrt{(k_a + k_d + k_i)^2 - 4k_d k_i} \right\rvert$, m$^3$·(mol·s)$^{-1}$ |
| $k_F$ | A function of second–order forward rate constants when F is inactivated in a ping–pong reaction : $\left\lvert \sqrt{(k_a + k_d + k_i)^2 - 4k_a k_i} \right\rvert$, m$^3$·(mol·s)$^{-1}$ |
| $k_S$ | A function of second–order forward rate constants when E is inactivated in a uni–uni reaction : $k_a k_c/(k_b + k_c) + k_i$, m$^3$·(mol·s)$^{-1}$ |
| Subscripts | |
| E | Enzyme state inactivated by suicide substrate |
| F | Intermediate enzyme state inactivated by suicide substrate |
| max | Maximum |
| (n)ss | (Non)quasi-steady state |
| n | Grade of the polynomial of a power expansion series |
| ssi | Suicide substrate inactivation |
| 0 | Condition at $t = 0$ |
| ∞ | Condition at $t \to \infty$ |

The kinetic analysis of the enzyme inactivation by suicide substrate was later expanded to the bi–bi Michaelis–Menten model, in which one of the two substrates was the inactivator [32]. Using the analytical method to solve the transient phase of enzyme systems [22,33], Varon et al. [32] established that the catalytic pathway of the bi–bi enzyme mechanisms remained in the steady state when the catalytic turnover between the competing catalytic and inactivation rate constants was much higher than the unit, however not when the rate constants were very close to each other. Additionally, the catalytic turnover,

in conjunction with the initial concentration of the enzyme, was employed to discriminate between different types of bi–bi enzyme mechanisms.

In the above two-step suicide substrate inactivation mechanisms with one or two substrates, the inactivator and the enzyme initially formed a reversible substrate–enzyme complex before, eventually, the enzyme irreversibly became inactive. However, there are situations in which the kinetics of the suicide substrate inactivation can be described as a one-step mechanism similar to that for more generic inactivators that irreversibly modify some specific amino acid residues with essential roles in the enzyme active site [6]. In this case, the one-step kinetic mechanism of the enzyme inactivation follows an irreversible second-order reaction that does not reach saturation by the suicide substrate. Excellent examples of the one-step suicide inactivation mechanism are the inactivation of peroxidase and catalase by $H_2O_2$ [34–36], in which $H_2O_2$ particularly inactivates an intermediate state of the enzyme formed during the catalytic turnover of a ping-pong reaction consisting of two stages. When $H_2O_2$ inactivation occurs, the catalytic and suicide inactivation reactions are presented as two concurrent irreversible second-order reactions. In particular, for the approximate solution of the nonlinear ODE system of the suicide inactivation of peroxidase by $H_2O_2$, it is required to fix conditions such that the concentration of $H_2O_2$ should both be constant during the time course of the reaction and higher than that of the nonsaturating concentration of the electron donor (or second substrate) [35,36]. In contrast, the kinetic analysis of the suicide inactivation of catalase by $H_2O_2$ does not introduce any restriction for the concentration of the $H_2O_2$, and none of the two concurrent second-order reactions reaches saturation by $H_2O_2$ [34,37]. The latter one-step suicide substrate inactivation with one substrate is generally schematized as shown below [34,38], where A is $H_2O_2$ and E stands for the total active catalase with no distinction between the initial and intermediate enzyme states of the ping-pong reaction.

$$A + E \xrightarrow{k_{a,\,cat}} E + P, \tag{4}$$

$$A + E \xrightarrow{k_{i,\,cat}} I. \tag{5}$$

This simplified reaction scheme has been extensively used with success to determine the overall inactivation rate constant of catalase when the enzyme was exposed to prolonged incubation with $H_2O_2$ under different experimental conditions such as temperature and pH [38–40]. More recently, a theoretical analysis of the inactivation reaction was conducted to design a rapid and high-throughput measurement of catalase in vitro [41].

The theoretical approach here presented aims to gain deeper insight into the kinetics leading to one-step suicide substrate inactivation of a ping-pong reaction in which the substrate undergoes disproportionation and can either irreversibly inactivate the initial or the intermediate active state of the enzyme, a situation not addressed in the above enzyme inactivation scheme (Reactions (4) and (5)). To achieve this goal, the standard quasi-steady-state and reactant stationary approximations [24] were examined initially in a uni–uni reaction following the irreversible Michaelis–Menten model, and then in a ping-pong reaction with one substrate undergoing disproportionation in the presence and absence of one-step suicide substrate inactivation. Despite the complexity of the ODE system of the enzyme-catalyzed ping–pong reaction in the presence of suicide substrate inactivation, the exploratory analysis of the approximate analytically integrated solutions was shown to be of relevance to identify the enzyme state that irreversibly reacted with the suicide substrate. Scheme 1 shows the flow to achieve the approximate analytically integrated solutions of the ODE systems of enzyme reactions under investigation in the present study.

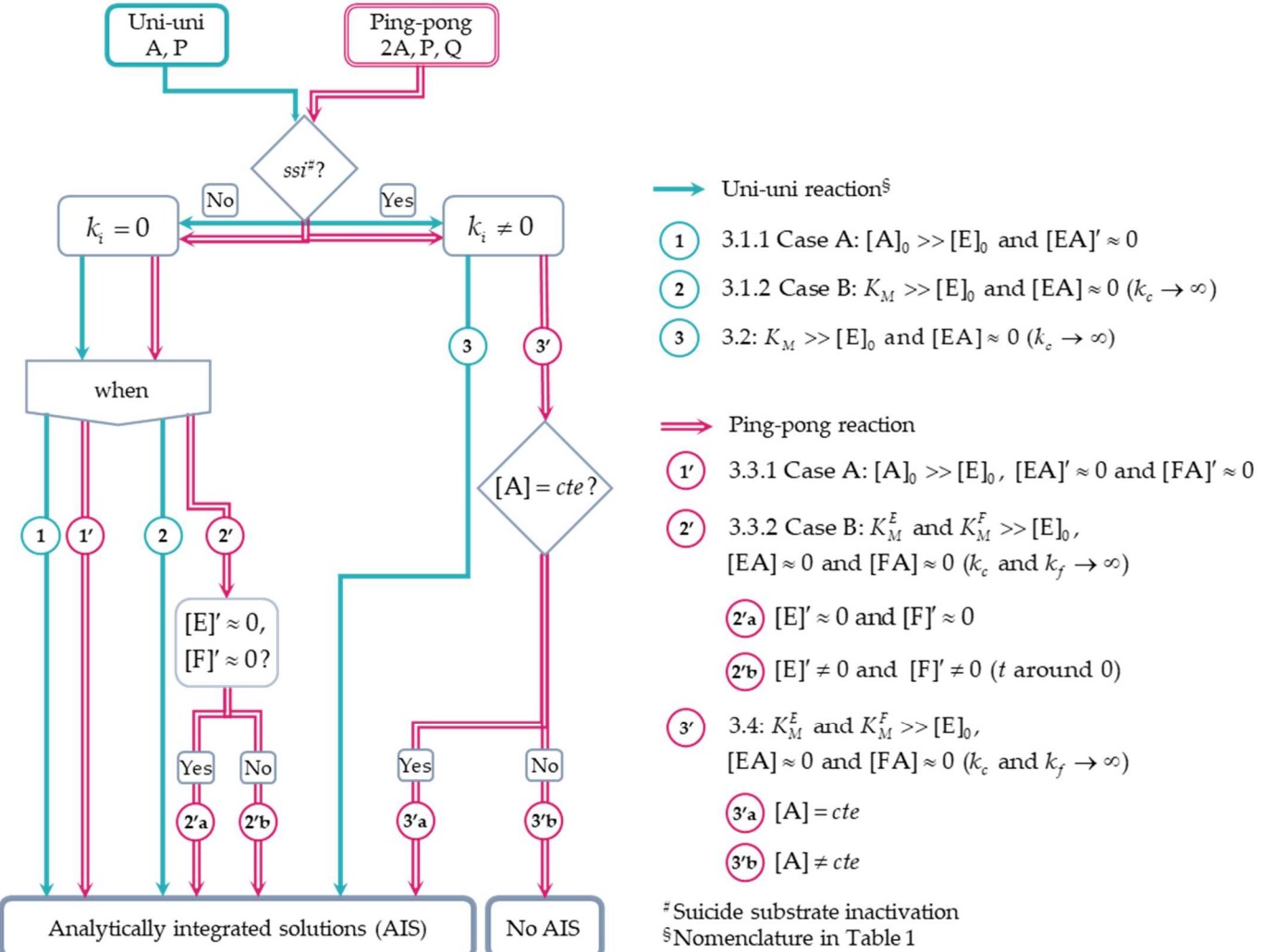

**Scheme 1.** Flow of the approximations used to solve the ODE system of the uni–uni and ping-pong reactions in the absence and presence of one-step substrate suicide inactivation. The numbers in the diagram correspond with the (sub)headings of Section 3.

## 2. Materials and Methods

The computer algebra system Wolfram Mathematica v. 12.2 (Champaign, IL, USA) [42] was used to program scripts to numerically and analytically solve the ODE systems of the irreversible uni–uni Michaelis–Menten and ping-pong reaction mechanisms, in which the enzyme followed a one-step suicide substrate inactivation. The Runge–Kutta method was applied to numerically solve the ODE system. The NDSolve and DSolve commands, together with other graphical and table commands, were programmed in Wolfram Mathematica in a manner such that the time and the values for initial concentrations and kinetic rate constants in the script could arbitrarily be modified. The values for the initial concentrations of the participating compounds and the kinetic rate constants were chosen to highlight key features of the enzyme reaction models in the graphical representations of the numerical and analytical solutions of the ODE systems. Thus, the molecular ratios between substrate and enzyme might not be representative of standard enzyme activity methods used in experimental laboratories. The approximate analytically integrated solutions of the time-dependent variation of the concentration for substrate, products, and enzyme were obtained after evaluating the standard quasi-steady-state and reactant stationary approximations [24]. The Taylor expansion series as the time approached zero were used when the DSolve command failed to provide an explicit analytical solution for the ODE

system under evaluation. The Lambert function is symbolized with the letter *W*. This function satisfies the equation $W(x) \times Exp[W(x)] = x$. The Wolfram Mathematica scripts for the analysis of the numerical and analytical solutions of the linear and nonlinear ODE systems and the graphical representations of the enzyme reaction models are available as Wolfram notebooks (Supplementary Materials, Figures S1A and S2A,B).

## 3. Results and Discussion

*3.1. Approximate Analytical Solutions for the Irreversible Uni–Uni Michaelis–Menten Model in the Absence of Suicide Substrate Inactivation*

The most common irreversible enzyme-catalyzed reaction involving one substrate and one substrate–enzyme complex is usually schematized as shown in Section 1 (Reaction (1)). The time-dependent variation of the reaction rate for each compound, using the mass action law and the compound stoichiometry, can be described with the corresponding nonlinear first-order ODEs as follows:

$$[A]' = -k_a[A][E] + k_b[EA], \tag{6}$$

$$[E]' = -k_a[A][E] + (k_b + k_c)[EA], \tag{7}$$

$$[EA]' = k_a[A][E] - (k_b + k_c)[EA], \tag{8}$$

$$[P]' = k_c[EA]. \tag{9}$$

Likewise, the following relations for concentrations and rates hold for reaction one:

$$[A]_0 = [A] + [EA] + [P], \tag{10}$$

$$[E]_0 = [E] + [EA], \tag{11}$$

$$0 = [A]' + [EA]' + [P]', \tag{12}$$

$$0 = [E]' + [EA]'. \tag{13}$$

3.1.1. Case A: $[A]_0 >> [E]_0$, $[EA]' \approx 0$

Figure 1A shows a representative numerical solution of the time dependence of the reaction rate for all the participating compounds in reaction one under conditions in which the standard quasi-steady-state approximation for the substrate–enzyme complex was not invoked. The maximum rate for $[P]$ in non-steady-state conditions (i.e., $[P]'_{max,\ nss}$) occurs when $[E]' = [EA]' = 0$. At this singular time, $-[A]' = [P]'_{max,\ nss}$, and the following, the relation is derived after substitution in Equation (9):

$$-[A]' = [P]'_{max,\ nss} = \frac{k_c[E]_0[A]}{(k_b + k_c)/k_a + [A]}. \tag{14}$$

Regardless of the initial values of the concentration for $[A]_0$ and $[E]_0$, or the finite values for the rate constants $k_a$, $k_b$, and $k_c$, Equation (14) shows that $[P]'_{max,\ nss}$ also matches the maximum rate under the conditions in which $[EA]$ is assumed to reach a steady state in the Michaelis–Menten equation. However, the match between $-[A]' = [P]'_{max,\ ss}$ in the Michaelis–Menten equation is imposed to go beyond one singular time. Equation (14) is valid for a reasonably prolonged time domain only if $[EA]$ is assumed to be in a quasi-steady state (i.e., $[EA]' \approx 0$) [17,18] after a rapid equilibrium of $[EA]$ with $[A]$ and $[E]$. The rapid equilibrium implies that $k_a \approx k_b > k_c$ [7]. However, the condition for a rapid equilibrium is not sufficient to derive the integrated Michaelis–Menten equation [15,21]. Thus, $[A]$ is not necessarily equal to $[A]_0$ at the time in which the quasi-steady state is reached unless the reactant stationary approximation is also satisfied [24]. This implies that the decrease in the substrate should be negligible during the transient phase, e.g., when $[A]_0 >> [E]_0$, a condition that is easily reached in most enzyme activity assays developed in experimental laboratories. If the second condition is fulfilled, the transient

phase is notably shortened [17,18]; therefore, the starting point of the quasi-steady-state approximation approaches the initial time. When this occurs, the shadowed area between $-[A]'$ and $[P]'$ in Figure 1A shrinks, and the two rates finally match.

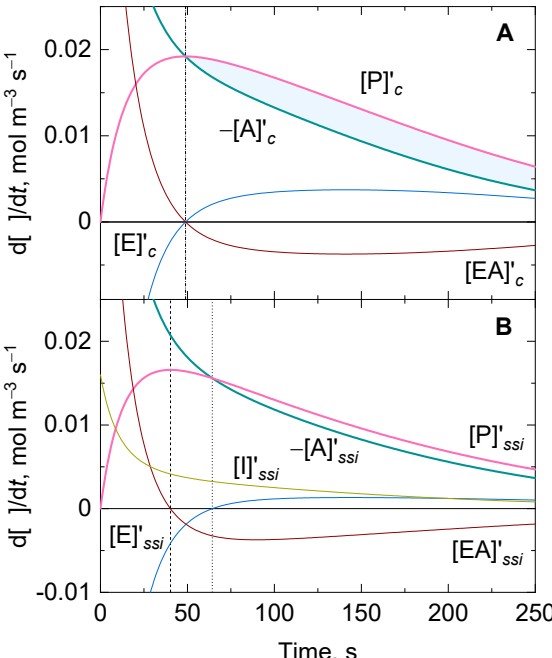

**Figure 1.** Representative numerical solutions of the time-dependent variation of the reaction rate for the participating compounds of an enzymatic system with one substrate and one substrate–enzyme complex in (**A**) the absence (subscript *c*) and (**B**) the presence of one-step suicide substrate inactivation (subscript *ssi*) under non-steady-state conditions. Vertical dashed and dotted lines show the time at which $[EA]'$ and $[E]'$ are equal to zero, respectively. The shadowed area between $-[A]'$ and $[P]'$ in (**A**) shows the gap that should be closed between these two rates to reach the steady state. Initial conditions: $[A]_0 = 4 \text{ mol} \cdot \text{m}^{-3}$, $[E]_0 = 2 \text{ mol} \cdot \text{m}^{-3}$, $k_a = 10^{-2} \text{ m}^3 \cdot (\text{mol} \cdot \text{s})^{-1}$, $k_b = 5 \times 10^{-3} \text{ s}^{-1}$, $k_c = 2 \times 10^{-2} \text{ s}^{-1}$, and $k_i = 2 \times 10^{-3} \text{ m}^3 \cdot (\text{mol} \cdot \text{s})^{-1}$.

Using the initial conditions $[A] = [A]_0$ and $[E] = [E]_0$, along with the reactant stationary approximation [24], a closed form of the integrated Michaelis–Menten equation can, thus, be represented as follows [43], in which $K_M$ is the Michaelis–Menten constant, and $W[\ ]$ is the Lambert function:

$$[A] = K_M \times W\left[\frac{[A]_0}{K_M} Exp\left(\frac{-k_c[E]_0 t + [A]_0}{K_M}\right)\right]. \tag{15}$$

### 3.1.2. Case B: $K_M >> [E]_0$, $[EA] \approx 0$

An alternative integrated solution, for which there is also a prolonged overlap between $-[A]'$ and $[P]'$, can be achieved when $K_M + [A]_0 >> [E]_0$, regardless of whether $[A]_0 >> [E]_0$ or $[A]_0 \approx [E]_0$ [24], a condition that also satisfies the validity of the total quasi-steady-state approximation [17]. For situations in which $[A]_0 \approx [E]_0$, the condition $K_M >> [E]_0$ can be fulfilled when $k_a \approx k_c < k_b$ or $k_a \approx k_b < k_c$. The ratio $k_a \approx k_c < k_b$, $[EA]$ accumulates and remains in a quasi-steady state for a prolonged time domain. In contrast, for the case in which $k_a \approx k_b < k_c$, $[EA]$ is negligible; hence, the substrate–enzyme complex does not accumulate significantly at any phase if $k_c \to \infty$ (i.e., $[EA] \approx 0$ and $[E] \approx [E]_0$). When this latter case occurs, the scheme of reaction one can be simplified to a single second-order reaction.

$$A + E \xrightarrow{k_a} E + P, \text{ if } k_c \to \infty. \tag{16}$$

Using the initial conditions $[A] = [A]_0$ and $[E] = [E]_0$, the approximate integrated solution is then:

$$[A] = [A]_0 e^{-k_a [E]_0 t}. \tag{17}$$

It is worth noting that this second integrated solution could also have been achieved if the limit of Equation (14) had been determined when $k_c$ approached infinity.

In brief, it can be established that the ODE system of the irreversible uni–uni Michaelis–Menten model can have approximate analytically integrated solutions under the reactant stationary approximation when either $[EA]' \approx 0$ or $[EA] \approx 0$. The second integrated solution is apparently an oversimplification of the irreversible enzyme-catalyzed reaction with one substrate–enzyme complex because, after all, it makes negligible the formation of the substrate–enzyme complex and the presence of the transient phase. However, the latter approximation cannot be ignored, as shown below, in enzyme-catalyzed reactions in which there is one-step suicide substrate inactivation, and an approximate analytically integrated solution is sought for the ODE system.

### 3.2. Approximate Analytical Solution for the Irreversible Uni–Uni Michaelis–Menten Model in the Presence of Suicide Substrate Inactivation

If the substrate is responsible for suicide inactivation, and the inactivation follows a one-step suicide mechanism, the new reaction to be added to the enzymatic scheme in reaction one is as follows:

$$A + E \xrightarrow{k_i} I. \tag{18}$$

With this new reaction, the ODE system of the uni–uni Michaelis–Menten model needs to be reformulated as follows:

$$[A]' = -(k_a + k_i)[A][E] + k_b[EA], \tag{19}$$

$$[E]' = -(k_a + k_i)[A][E] + (k_b + k_c)[EA], \tag{20}$$

$$[EA]' = k_a[A][E] - (k_b + k_c)[EA], \tag{21}$$

$$[P]' = k_c[EA], \tag{22}$$

$$[I]' = k_i[A][E]. \tag{23}$$

For the new ODE system, the following relations for concentrations and rates also hold:

$$[A]_0 = [A] + [EA] + [P] + [I], \tag{24}$$

$$[E]_0 = [E] + [EA] + [I], \tag{25}$$

$$0 = [A]' + [EA]' + [P]' + [I]', \tag{26}$$

$$0 = [E]' + [EA]' + [I]'. \tag{27}$$

Figure 1B shows a representative time-dependent variation of the reaction rate for all of the compounds when there is one-step suicide substrate inactivation. When $k_i > 0$, the active enzyme is irreversibly consumed. Reaction (18) does not reach saturation by the inactivator and, strictly speaking, there is no condition for which $[EA]$ remains in a quasi-steady state during a prolonged time domain. At $t = 0$, $[EA] = 0$, and it increases until the time at which $[EA]' = 0$. Beyond this instance, the value for $[EA]$ decreases and approaches zero, regardless of whether $[A]$ remains high and finally $[I]_\infty = [E]_0$ or $[A]$ is exhausted and $[I]_\infty = [E]_0 - [E]_\infty$. Additionally, when $k_i$ is small, there are also conditions for which $[E]$ can reach a minimum (i.e., $[E]' = 0$), and thus, $-[A]' = [P]'$. However, $[P]'$ is not maximum at the time at which $[E]' = 0$. In contrast, if $k_i$ is high, there might be no time at which $[E]' = 0$, while the enzyme is still active and, hence, there will be no condition for which $-[A]' = [P]'$ (data not shown in Figure 1B).

The time dependence of $[P]'$ shows that it reaches its maximum at $[EA]' = 0$, regardless of the value for $k_i$. Bearing this in mind, the following equations can be derived from the ODE system containing Equations (19)–(23):

$$[A]' = -\left(\frac{k_a k_c}{k_b + k_c} + k_i\right)[A][E], \tag{28}$$

$$[E]' = -k_i[A][E], \tag{29}$$

$$[P]'_{max,nss} = -[A]' + [E]'. \tag{30}$$

Once again, one must keep in mind that these rates (Equations (28)–(30)) are only valid for the singular time at which $[EA]' = 0$. If attempts are now conducted to find conditions for which the above reaction rates for $[A]$ and $[E]$ are expected to be valid for a prolonged time domain, one has to inspect the overlap of some representative numerically and analytically integrated solutions of the ODE system by changing the values for the initial concentrations and rate constants. Accepting $[A] = [A]_0$ and $[E] = [E]_0$ as the starting points for integration, the following integrated equations for $[A]$, $[E]$, $[P]$, and $[I]$ were obtained:

$$[A] = \frac{[A]_0(k_i[A]_0 - k_S[E]_0)}{k_i[A]_0 - k_S[E]_0 \times \text{Exp}[-(k_i[A]_0 - k_S[E]_0)t]}, \tag{31}$$

$$[E] = \frac{[E]_0(k_i[A]_0 - k_S[E]_0)}{k_i[A]_0 \times \text{Exp}[(k_i[A]_0 - k_S[E]_0)t] - k_S[E]_0}, \tag{32}$$

$$[P] = [A]_0 - [A] + [E] - [E]_0, \tag{33}$$

$$[I] = [E]_0 - [E], \tag{34}$$

where:

$$k_S = k_a k_c / (k_b + k_c) + k_i. \tag{35}$$

After trial and error visualization, it was possible to confirm there was no overlap between the numerically and analytically integrated solutions of the time-dependent variation of the concentration for the participating compounds when arbitrarily varying the values for $[A]_0$, $[E]_0$, and the rate constants. This clearly indicates that other conditions should be imposed on the initial concentrations, or the rate constants of the enzymatic system to find the expected overlap between the numerically and analytically integrated solutions if Equations (31)–(34) were to be valid for a prolonged time domain. The application of the reactant stationary approximation, $K_M >> [E]_0$, for the case in which $k_a \approx k_b < k_c$ (i.e., $k_c \to \infty$), simplifies $k_S$ to the sum of the two second-order rate constants of the two competing reactions for the suicide substrate (i.e., $k_S \approx k_a + k_i$). Now, the respective numerically and analytically integrated solutions for $[A]$, $[E]$, $[P]$, and $[I]$ are merged, and their overlap remains regardless of the values for $[A]_0$ and $[E]_0$, and for $k_a$ and $k_i$. These approximate analytically integrated solutions are equivalent to those shown in [34,39,41] for the one-step suicide inactivation of catalase by $H_2O_2$, in which the kinetic mechanism analysis does not consider the formation of the intermediate active state of the enzyme. Therefore, when there is one-step suicide substrate inactivation in an irreversible uni–uni Michaelis–Menten model, it can be established that the ODE system of the reaction scheme can be analytically solved if the approximation $[EA] \approx 0$ is accepted in any time domain in which the enzyme reaction takes place (Figure 2).

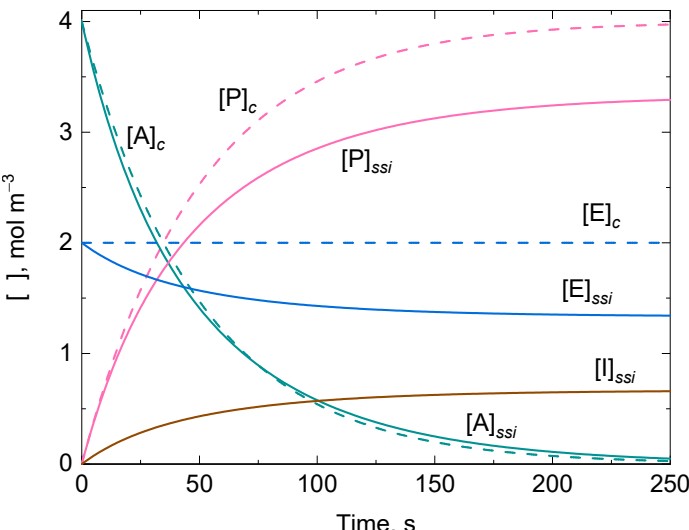

**Figure 2.** Representative approximate analytically integrated solutions of the time-dependent variation of the concentration for the participating compounds of an irreversible uni–uni Michaelis–Menten model in the absence (dashed lines, subscript *c*) and presence of suicide substrate inactivation (solid lines, subscript *ssi*) following a one-step mechanism. Initial conditions: $[A]_0 = 4$ mol $\cdot$m$^{-3}$, $[E]_0 = 2$ mol $\cdot$m$^{-3}$, $k_a = 10^{-2}$ m$^3 \cdot$(mol$\cdot$ s)$^{-1}$, and $k_i = 2 \times 10^{-3}$ m$^3 \cdot$(mol$\cdot$ s)$^{-1}$.

### 3.3. Analytical Solutions for an Enzyme-Catalyzed Ping-Pong Reaction with One Substrate Undergoing Disproportionation in the Absence of Suicide Substrate Inactivation

An enzymatic reaction, in which the substrates and products alternatively bind to and leave out the enzyme, is said to follow a ping-pong reaction. In this type of enzyme-catalyzed reaction, there are occasions in which one single substrate undergoes disproportionation (i.e., usually an enzymatic redox transformation in which the same substrate can be both oxidized and reduced, and then be converted to two or more different products). The substrate, thus, binds to the enzyme at two different stages per catalytic turnover. Two well-known examples in enzymology are represented by the disproportionation of the superoxide radical ($O_2^{\bullet-}$) and $H_2O_2$ by superoxide dismutase and catalase enzymes, respectively [37,44]. The enzyme-catalyzed disproportionation reaction here presented differs from other uni–bi Michaelis–Menten models in which the substrate binds to the enzyme only once per catalytic turnover, however two substrate–enzyme complexes are involved, each forming a different product [45,46]. Thus, the ping-pong reaction with a substrate following disproportionation can be represented as follows:

$$A + E \underset{k_b}{\overset{k_a}{\rightleftarrows}} EA \overset{k_c}{\rightarrow} F + P, \tag{36}$$

$$A + F \underset{k_e}{\overset{k_d}{\rightleftarrows}} FA \overset{k_f}{\rightarrow} E + Q. \tag{37}$$

In the first stage, the substrate reversibly forms a substrate–enzyme complex EA with the free enzyme. After the irreversible release of the first product, the enzyme remains in an active intermediate state that reversibly binds to the substrate in the second stage. The new substrate–enzyme complex FA is different from the previous one and produces a second product that eventually irreversibly leaves out the active site of the enzyme. Finally, the enzyme returns to its initial state, and the reaction cycle repeats until the substrate is fully consumed.

The ODE system of the time-dependent variation of the reaction rate for the participating compounds of this ping-pong reaction (Reactions (36) and (37)) is as follows:

$$[A]' = -(k_a[E] + k_d[F])[A] + (k_b[EA] + k_e[FA]), \tag{38}$$

$$[E]' = -k_a[A][E] + (k_b[EA] + k_f[FA]), \tag{39}$$

$$[EA]' = k_a[A][E] - (k_b + k_c)[EA], \tag{40}$$

$$[F]' = -k_d[A][F] + (k_c[EA] + k_e[FA]), \tag{41}$$

$$[FA]' = k_d[A][F] - (k_e + k_f)[FA], \tag{42}$$

$$[P]' = k_c[EA], \tag{43}$$

$$[Q]' = k_f[FA]. \tag{44}$$

For this ODE system, the following relations for concentrations and rates also hold:

$$[A]_0 = [A] + [EA] + [FA] + [P] + [Q], \tag{45}$$

$$[E]_0 = [E] + [F] + [EA] + [FA], \tag{46}$$

$$0 = [A]' + [EA]' + [FA]' + [P]' + [Q]', \tag{47}$$

$$0 = [E]' + [F]' + [EA]' + [FA]'. \tag{48}$$

For conditions in which the substrate–enzyme complexes are not in a quasi-steady state, there is a time $t$ for which the decomposition rate of the substrate matches the sum of the formation rates of the products (Figure 3).

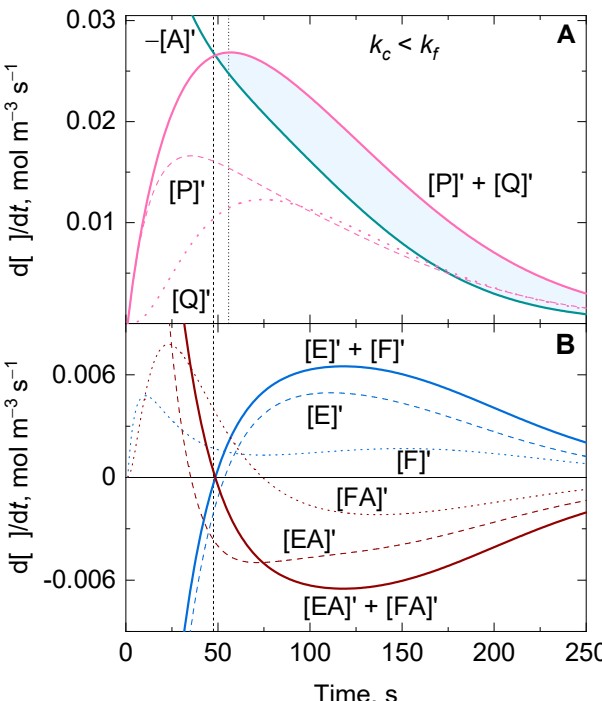

**Figure 3.** Representative numerical solutions of the time-dependent variation of the reaction rate for (**A**) the substrate A, and the products P and Q, and (**B**) the enzyme states E and F, and the substrate–enzyme complexes EA and FA of an enzyme-catalyzed ping-pong reaction with one substrate undergoing disproportionation under non-steady-state conditions. Vertical dashed and dotted lines show the time at which $[EA]' + [FA]' = 0$ and $k_c[EA]' + k_f[FA]' = 0$, respectively. The shadowed area between $-[A]'$ and $[P]' + [Q]'$ in (**A**) shows the gap that should be closed between the substrate and product rates to reach the steady-state conditions. Initial conditions: $[A]_0 = 4 \text{ mol} \cdot \text{m}^{-3}$, $[E]_0 = 2 \text{ mol} \cdot \text{m}^{-3}$, $k_a = 10^{-2} \text{ m}^3 \cdot (\text{mol} \cdot \text{s})^{-1}$, $k_b = 5 \times 10^{-3} \text{ s}^{-1}$, $k_c = 2 \times 10^{-2} \text{ s}^{-1}$, $k_d = 5 \times 10^{-2} \text{ m}^3 \cdot (\text{mol} \cdot \text{s})^{-1}$, $k_e = 4 \times 10^{-3} \text{ s}^{-1}$, and $k_f = 4 \times 10^{-2} \text{ s}^{-1}$.

At this singular time, the sum of the concentrations of the two substrate–enzyme complexes reaches its maximum (i.e., $[EA]' + [FA]' = 0$), and, hence, $-[A]' = [P]' + [Q]'$.

However, in contrast to the reaction scheme of the irreversible uni–uni Michaelis–Menten model with one single substrate–enzyme complex, the former condition does not necessarily imply that the sum of the product rates is also maximum at that time. In fact, a closer inspection of the time dependence of the reaction rate for the participating compounds shows that, firstly, the maximum of $[P]' + [Q]'$ is reached at the time at which the sum $k_c[EA] + k_f[FA]$ is maximum (or $k_c[EA]' + k_f[FA]' = 0$), and, secondly, only if, particularly for $k_c = k_f$, there is a time at which the equality $-[A]' = ([P]' + [Q]')_{\text{max, } nss}$ also holds. The maximum of the sum of the product rates $(k_c[EA] + k_f[FA])$ can appear before or after the time at which $[EA]' + [FA]' = 0$, and this simply depends on whether the ratio between $k_c$ and $k_f$ is higher or lower than the unit. Although all these relations might provide a hint about the ping-pong reaction in which the substrate undergoes disproportionation under non-steady-state conditions, the ODE system still remains very complex, and $-[A]' = ([P]' + [Q]')_{\text{max, } nss}$ cannot be expressed as an analytical function that only depends on the concentrations of $[E]_0$ and $[A]$, as shown for the irreversible uni–uni Michaelis–Menten model with only one substrate–enzyme complex (Equation (14)).

3.3.1. Case A: $[A]_0 >> [E]_0$, $[EA]' \approx 0$, $[FA]' \approx 0$

In order to reach an analytical function in which $-[A]'$ only depends on $[E]_0$ and $[A]$, yet not on the enzyme intermediates, it is necessary to invoke rapid equilibria of $[EA]$ with $[A]$ and $[E]$, and of $[FA]$ with $[A]$ and $[F]$ to fulfill the condition that both $[EA]$ and $[FA]$ are in a quasi-steady state (i.e., $[EA]' \approx 0$ and $[FA]' \approx 0$). If steady-state conditions are thus applied [7,47], the analytical expression for $-[A]'$ can be achieved following the King–Altman method [48], while keeping in mind that the following equalities between reaction rates for the participating compounds must also hold due to the stoichiometry of Reactions (36) and (37):

$$-\frac{1}{2}[A]' = [P]' = [Q]'. \tag{49}$$

The enzyme reaction rate is, thus, as follows:

$$-\frac{1}{2}[A]' = [P]'_{\text{max, } ss} = [Q]'_{\text{max, } ss} = \frac{k_c k_f [E]_0}{k_c + k_f} \frac{[A]}{K_M^E + K_M^F + [A]}, \tag{50}$$

where:

$$K_M^E = k_f(k_b + k_c) / \left[ k_a \left( k_c + k_f \right) \right], \tag{51}$$

$$K_M^F = k_c \left( k_e + k_f \right) / \left[ k_d \left( k_c + k_f \right) \right]. \tag{52}$$

Equation (50) shows that $-[A]' = 2[P]'_{\text{max, } ss}$ or $-[A]' = 2[Q]'_{\text{max, } ss}$. However, in contrast to the first case for one substrate–enzyme complex (Equation (14)), the maxima of the sum of the product rates under non-steady-state and quasi-steady-state conditions are not coincident [i.e., $([P]' + [Q]')_{\text{max, } nss} \neq 2[P]'_{\text{max, } ss}$ or $([P]' + [Q]')_{\text{max, } nss} \neq 2[Q]'_{\text{max, } ss}$]. In other words, there is no time, except for a singular case, at which $([P]' + [Q]')_{\text{max, } nss}$ matches the value for $2[P]'_{\text{max, } ss}$ or $2[Q]'_{\text{max, } ss}$, regardless of the chosen initial values for $[A]_0$, $[E]_0$ and the rate constants. The singular case can only be found if the counterpart rate constants of the two stages of the ping-pong reaction were set equal to each other (i.e., $k_a = k_d$, $k_b = k_e$, and $k_c = k_f$). These matches between rate constants are not expected to be found under experimental conditions and, in the event that they are, the enzyme mechanism can simply be treated as an enzyme reaction following the irreversible uni–uni Michaelis–Menten model. As stated for case A in Section 3.1.1, the condition for rapid equilibrium is not sufficient to derive the integrated Michaelis–Menten equation. Thus, $[A]$ is not necessarily equal to $[A]_0$ in the transient phase unless the reactant stationary

approximation is also satisfied [24]. Therefore, Equation (50) can be integrated using $[A] = [A]_0$ and $[E] = [E]_0$ if $[A]_0 >> [E]_0$:

$$\frac{2k_c k_f [E]_0 t}{\left(k_c + k_f\right)} = \left(K_M^E + K_M^F\right) \ln \frac{[A]_0}{[A]} + [A]_0 - [A]. \tag{53}$$

A closed-form solution for the time-dependent variation of $[A]$ is also given in the Supplementary Materials.

3.3.2. Case B: $K_M^E$ and $K_M^F >> [E]_0$, $[EA] \approx 0$, $[FA] \approx 0$

Similar to Case B in Section 3.1.2, another solution for which there will also be a prolonged overlap between $-[A]'$ and $[P]' + [Q]'$ can be obtained if both $K_M^E$ and $K_M^F >> [E]_0$, regardless of whether $[A]_0 >> [E]_0$ or $[A]_0 \approx [E]_0$. The reasoning follows that given for the irreversible uni–uni Michaelis–Menten model. The conditions $K_M^E >> [E]_0$ and $K_M^F >> [E]_0$ can be fulfilled when $k_a \approx k_b < k_c$ and $k_d \approx k_e < k_f$, respectively. This now implies that both $[EA]$ and $[FA]$ are negligible and do not accumulate at any phase; hence, $[E]_0 \approx [E] + [F]$ if $k_c \to \infty$ and $k_f \to \infty$. When this occurs, the scheme of reactions (36) and (37) can be simplified to two single second-order reactions as follows:

$$A + E \xrightarrow{k_a} F + P, \tag{54}$$

$$A + F \xrightarrow{k_d} E + Q. \tag{55}$$

For the two new enzyme reactions, the following relations for concentrations and reaction rates also hold:

$$[A]_0 = [A] + [P] + [Q], \tag{56}$$

$$[E]_0 = [E] + [F], \tag{57}$$

$$0 = [A]' + [P]' + [Q]', \tag{58}$$

$$0 = [E]' + [F]'. \tag{59}$$

Thus, the nonlinear ODE system of the time-dependent variation of the reaction rate for the participating compounds can be reduced to the following two differential equations:

$$[A]' = -[(k_a - k_d)[E] + k_d[E]_0][A], \tag{60}$$

$$[E]' = -[(k_a + k_d)[E] - k_d[E]_0][A]. \tag{61}$$

To solve Equations (60) and (61), the chain rule $dz/dx = dz/dy \cdot dy/dx$ was applied using $[A] = [A]_0$ and $[E] = [E]_0$ as the integration starting points to determine the enzyme-dependent variation of $[A]$:

$$[A] = [A]_0 + \frac{k_a - k_d}{k_a + k_d}([E] - [E]_0) + \frac{2k_a k_d [E]_0}{(k_a + k_d)^2} \ln\left[\frac{(k_a + k_d)[E] - k_d[E]_0}{k_a[E]_0}\right]. \tag{62}$$

Equation (62) is of interest because it can directly be used to determine the final concentration of $[E]_\infty$ and $[F]_\infty$, when $[A]$ becomes exhausted (i.e., $[A] = 0$) at the end of the reaction, regardless of the initial value for $[A]_0$ (Figure S2).

$$[E] = \frac{k_d \left[k_a - k_d + 2k_a W\left[\frac{(k_a - k_d)}{2k_d} \text{Exp}\left[\frac{k_a(k_a - k_d)[E]_0 - (k_a + k_d)^2[A]_0}{2k_a k_d [E]_0}\right]\right]\right]}{(k_a - k_d)(k_a + k_d)}[E]_0, \tag{63}$$

where the Lambert function (i.e., $W[\ ]$) rapidly approaches zero when $[A]_0 >> [E]_0$.

For a particular situation in which $[A]_0$ is in excess in comparison to $[E]_0$, the values for $[E]_\infty$ and $[F]_\infty$ do not depend on $[A]_0$, and they can be determined as the limit of $[E]$ when $[A]_0$ approaches infinity.

$$[E]_\infty = \lim_{[A]_0 \to \infty} [E] = \frac{k_d}{(k_a + k_d)} [E]_0, \tag{64}$$

$$[F]_\infty = [E]_0 - [E]_\infty = \frac{k_a}{(k_a + k_d)} [E]_0. \tag{65}$$

The next natural step ought to be the substitution of the enzyme-dependent function of $[A]$ (Equation (62)) for $[A]$ in $[E]'$ (Equation (61)), followed by the integration of the equation by separation of variables.

$$[E]' = -[(k_a + k_d)[E] - k_d[E]_0] \left[ [A]_0 + \frac{k_a - k_d}{k_a + k_d}([E] - [E]_0) + \frac{2k_a k_d [E]_0}{(k_a + k_d)^2} \ln\left[ \frac{(k_a + k_d)[E] - k_d[E]_0}{k_a[E]_0} \right] \right]. \tag{66}$$

However, it was found that there was no analytically integrated solution for $[E]'$ (Equation (66)). Therefore, alternatively, and in an attempt to find approximate analytically integrated solutions, two approaches were attempted. In the first approach, the nonlinear ODE system (Equations (60) and (61)) was easily solved when the steady-state approximation was invoked for $[E]$, and, consequently, for $[F]$. In the second approach, approximate solutions for $[A]$ and $[E]$ were found under non-steady-state conditions around $t = 0$. Both alternative approaches had some advantages, however disadvantages are also discussed with some more detail in the Supplementary Materials. For the special case in which $k_a = k_d$, the approximate analytically integrated solution for $[A]$ was equivalent to Equation (17); thus, it is not further discussed here.

Accordingly, the analytical and numerical solutions of the time dependence of $-[A]'$ and $[E]'$ can be divided into three domains of time as shown in Figure 4.

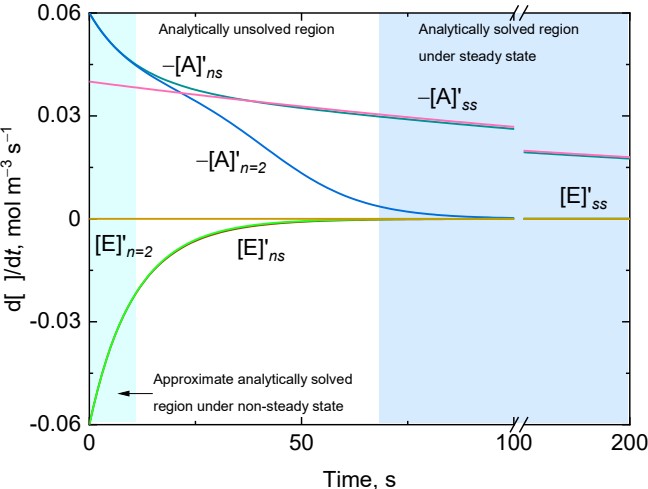

**Figure 4.** Representative numerical (subscript *ns*) and approximate analytical (subscript *n* = 2) solutions of the time-dependent variation of the reaction rate for the substrate A undergoing disproportionation and one of the active enzymes states, E, of an enzyme-catalyzed ping-pong reaction in which the substrate–enzyme complexes do not accumulate. The numerical solutions (*ns*) for $-[A]'$ and $[E]'$ are shown together with two approximate analytical solutions for $-[A]'$ and $[E]'$ in two different domains: the pale-green domain for which the steady-state approximations are not invoked around $t = 0$ (i.e., $-[A]'_{n=2}$ and $[E]'_{n=2}$) and the pale-blue domain for which the steady-state (*ss*) approximation is invoked (i.e., $-[A]'_{ss}$ and $[E]'_{ss}$). The intermediate non-shadowed domain is not analytically solved. The equality $-[E]' = [F]'$ holds for any domain. Initial conditions: $[A]_0 = 10 \text{ mol} \cdot \text{m}^{-3}$, $[E]_0 = 1 \text{ mol} \cdot \text{m}^{-3}$, $k_a = 6 \times 10^{-3} \text{ m}^3 \cdot (\text{mol} \cdot \text{s})^{-1}$, and $k_d = 3 \times 10^{-3} \text{ m}^3 \cdot (\text{mol} \cdot \text{s})^{-1}$.

The first domain around $t = 0$ respectively shows the match of the numerical solutions of $-[A]'$ and $[E]'$ with the approximate solutions for $-[A]'_{n=2}$ and $[E]'_{n=2}$ under non-steady-state conditions (see Supplementary Materials). This domain is followed by an intermediate domain for which there are no analytical solutions for $-[A]'$ and $[E]'$, and the mismatch between the numerical and analytical solutions was more prominent, particularly for $-[A]'$. The last domain corresponds with the region for which $[E]$ and $[F]$ were in a steady state and, hence, their rates approached zero (Equations (S2) and (S3)). In this last domain, the time-dependent variation of $[A]$ is an exponential function that can easily be expressed in terms of $[E]_0$. The latter analytical solution is widely used in the kinetic analysis of enzymes such as catalase when suicide substrate inactivation by $H_2O_2$ (at low concentration) is negligible, and steady-state conditions are considered to be rapidly reached [37].

### 3.4. Analytical Solutions for an Enzyme-Catalyzed Ping-Pong Reaction with One Substrate Undergoing Disproportionation in the Presence of Suicide Substrate Inactivation

The presence of suicide substrate inactivation in an enzyme reaction with two active states opens the question of which enzyme state is inactivated by the suicide substrate and whether it can be identified. Figure 5 shows representative numerically integrated solutions for the time-dependent variation of $[A]$ and the sum of $[P] + [Q]$ in a ping-pong reaction in which the substrate underwent disproportionation and steady-state conditions were not invoked. In the example, one-step suicide substrate inactivation occurred in the active state E or F following second-order reactions that did not reach saturation by the substrate (Reactions (69) and (78)). The formation of the intermediate substrate–enzyme complexes was considered negligible in Reactions (67) and (68), as well as in Reactions (76) and (77), although the conclusions here presented for the numerically integrated solutions did not depend on it. The plots undoubtedly show that the traces for $[A]$ and $[P] + [Q]$ vary notably depending on the enzyme state inactivated by the suicide substrate, implying that the mathematical expressions of their approximate analytically integrated solutions should also reflect such differences.

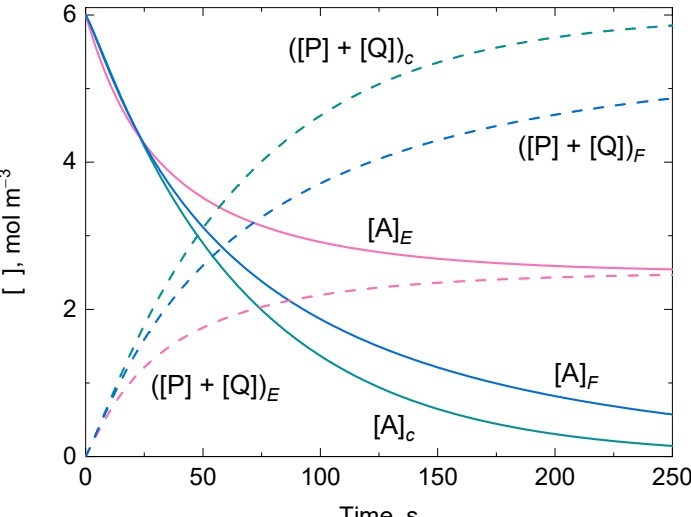

**Figure 5.** Representative numerically integrated solutions of the time-dependent variation of the concentration for the substrate A undergoing disproportionation, and the sum of products P and Q of an enzyme-catalyzed ping-pong reaction in which suicide substrate inactivation by A occurs when the enzyme is in the active state E (subscript *E*) or F (subscript *F*). The subscript *c* stands for control. Initial conditions $[A]_0 = 6$ mol $\cdot$m$^{-3}$, $[E]_0 = 1$ mol$\cdot$ m$^{-3}$, $k_a = 10^{-2}$ m$^3$(mol$\cdot$ s)$^{-1}$, $k_d = 3 \times 10^{-2}$ m$^3\cdot$(mol $\cdot$s)$^{-1}$, and $k_i = 8 \times 10^{-3}$ m$^3\cdot$(mol $\cdot$s)$^{-1}$. The value for $k_i$ was the same regardless of which enzyme state reacted with the suicide substrate.

Following the same reasoning given in Section 3.2 to find a match between the numerically and analytically integrated solutions of the time-dependent variation of the concentration for the participating compounds, the conditions $k_a \approx k_b < k_c$ and $k_d \approx k_e < k_f$ (i.e., $k_c \to \infty$ and $k_f \to \infty$) were imposed. This means that $K_M^E >> [E]_0$ and $K_M^F >> [E]_0$; hence, $[EA] \approx 0$ and $[FA] \approx 0$ The two potential inactivation mechanisms were, thus, schematized as follows:

3.4.1. Case A: Suicide Substrate Inactivation on E

For this first case, the enzymatic reaction can be represented by the following scheme:

$$A + E \xrightarrow{k_a} F + P, \tag{67}$$

$$A + F \xrightarrow{k_d} E + Q, \tag{68}$$

$$A + E \xrightarrow{k_i} I. \tag{69}$$

Thus, the time-dependent variation of the reaction rate for the participating compounds can be described as follows:

$$[A]' = -[(k_a + k_i)[E] + k_d[F]][A], \tag{70}$$

$$[F]' = (k_a[E] - k_d[F])[A], \tag{71}$$

$$[E]' = [k_d[F] - (k_a + k_i)[E]][A], \tag{72}$$

$$[P]' = k_a[E][A], \tag{73}$$

$$[Q]' = k_d[F][A], \tag{74}$$

$$[I]' = k_i[E][A]. \tag{75}$$

3.4.2. Case B: Suicide Substrate Inactivation on F

For this second case, the enzymatic reaction can be represented by the scheme:

$$A + E \xrightarrow{k_a} F + P, \tag{76}$$

$$A + F \xrightarrow{k_d} E + Q, \tag{77}$$

$$A + F \xrightarrow{k_i} I. \tag{78}$$

Thus, the time-dependent variation of the reaction rate for the participating compounds can be described as follows:

$$[A]' = -[k_a[E] + (k_d + k_i)[F]][A], \tag{79}$$

$$[F]' = [k_a[E] - (k_d + k_i)[F]][A], \tag{80}$$

$$[E]' = (k_d[F] - k_a[E])[A], \tag{81}$$

$$[P]' = k_a[E][A], \tag{82}$$

$$[Q]' = k_d[F][A], \tag{83}$$

$$[I]' = k_i[F][A]. \tag{84}$$

Attempts to solve either of the above two nonlinear ODE systems were infructuous. Therefore, to find approximate analytically integrated solutions in which the analysis could allow for distinguishing between E or F inactivation by suicide substrate, it was considered an enzyme reaction in which the substrate was continuously added into the reaction system and thus [A] remained constant and unaffected by time. If this is now the case, the ODE

system becomes linear, and the time-dependent variation of $[E]'$ and $[F]'$ can be solved and their integrated solutions be substituted for $[E]$ and $[F]$ in $[P]'$ and $[Q]'$ to finally obtain the approximate analytically integrated solutions of $[P]$ and $[Q]$.

For case A, in which the enzyme state E was inactivated by the substrate, the integrated solutions were as follows:

$$[E] = \frac{1}{2k_E}[E]_0 e^{-\frac{1}{2}(k_a+k_d+k_i+k_E)[A]_0 t}\left[(k_a - k_d + k_i)\left(1 - e^{k_E[A]_0 t}\right) + k_E\left(1 + e^{k_E[A]_0 t}\right)\right], \quad (85)$$

$$[F] = \frac{k_a}{k_E}[E]_0 e^{-\frac{1}{2}(k_a+k_d+k_i+k_E)[A]_0 t}\left(e^{k_E[A]_0 t} - 1\right), \quad (86)$$

$$[P] = \frac{k_a}{2k_i k_E}[E]_0 e^{-\frac{1}{2}(k_a+k_d+k_i+k_E)[A]_0 t}\left[(k_a + k_d - k_i)\left(1 - e^{k_E[A]_0 t}\right) - k_E\left(1 + e^{k_E[A]_0 t} - 2e^{\frac{1}{2}(k_a+k_d+k_i+k_E)[A]_0 t}\right)\right], \quad (87)$$

$$[Q] = \frac{k_a}{2k_i k_E}[E]_0 e^{-\frac{1}{2}(k_a+k_d+k_i+k_E)[A]_0 t}\left[(k_a + k_d + k_i)\left(1 - e^{k_E[A]_0 t}\right) - k_E\left(1 + e^{k_E[A]_0 t} - 2e^{\frac{1}{2}(k_a+k_d+k_i+k_E)[A]_0 t}\right)\right], \quad (88)$$

$$[I] = \frac{k_i}{k_a}[P], \quad (89)$$

where:

$$k_E = \left|\sqrt{(k_a + k_d + k_i)^2 - 4k_d k_i}\right|. \quad (90)$$

The integrated solutions for case B, in which the intermediate enzyme state F was inactivated by the substrate, were obtained following the same procedure:

$$[E] = \frac{1}{2k_F}[E]_0 e^{-\frac{1}{2}(k_a+k_d+k_i+k_F)[A]_0 t}\left[(k_a - k_d - k_i)\left(1 - e^{k_F[A]_0 t}\right) + k_F\left(1 + e^{k_F[A]_0 t}\right)\right], \quad (91)$$

$$[F] = \frac{k_a}{k_F}[E]_0 e^{-\frac{1}{2}(k_a+k_d+k_i+k_F)[A]_0 t}\left(e^{k_F[A]_0 t} - 1\right), \quad (92)$$

$$[P] = \\ \frac{1}{2k_i k_F}[E]_0 e^{-\frac{1}{2}(k_a+k_d+k_i+k_F)[A]_0 t} \times \\ \left[k_a(k_d - k_i)\left(1 - e^{k_F[A]_0 t}\right) + (k_d + k_i)\left((k_d + k_i)\left(1 - e^{k_F[A]_0 t}\right) - k_F\left(1 + e^{k_F[A]_0 t} - 2e^{\frac{1}{2}(k_a+k_d+k_i+k_F)[A]_0 t}\right)\right)\right], \quad (93)$$

$$[Q] = \frac{k_d}{2k_i k_F}[E]_0 e^{-\frac{1}{2}(k_a+k_d+k_i+k_F)[A]_0 t}\left[(k_a + k_d + k_i)\left(1 - e^{k_F[A]_0 t}\right) - k_F\left(1 + e^{k_F[A]_0 t} - 2e^{\frac{1}{2}(k_a+k_d+k_i+k_F)[A]_0 t}\right)\right], \quad (94)$$

$$[I] = \frac{k_i}{k_d}[Q], \quad (95)$$

where:

$$k_F = \left|\sqrt{(k_a + k_d + k_i)^2 - 4k_a k_i}\right|. \quad (96)$$

As time goes by, the enzyme is consumed regardless of which state is inactivated by the suicide substrate (Figure 6). The sum of concentrations $[E] + [F]$ becomes zero as $t$ approaches infinity, while the concentration of the inactivate state of the enzyme (i.e., I) increases until $[I]$ is equal to $[E]_0$. At the same time, the product formation ceases, and $[P]$ and $[Q]$ reach constant values in the reaction mixture. The values for $[P]$ and $[Q]$ can be evaluated at the end of the reaction if the limit of the integrated solutions is determined when $t$ approaches infinity.

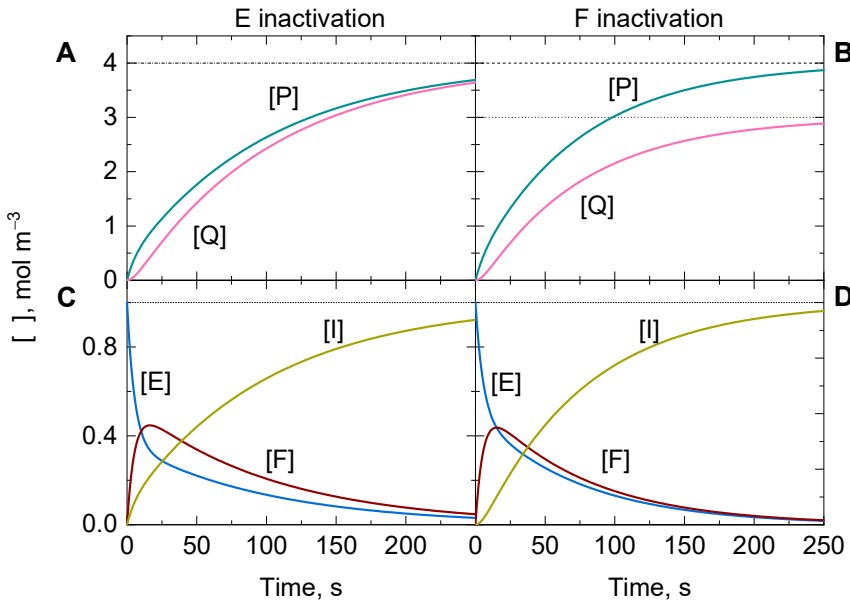

**Figure 6.** Representative approximate analytically integrated solutions of the time-dependent variation of the concentration for (**A**,**B**) the products P and Q and (**C**,**D**) the two active enzyme states E and F, together with the inactivated state I, of a ping-pong reaction under non-steady-state conditions, in which the suicide substrate A undergoes disproportionation and inactivates (**A**,**C**) E or (**B**,**D**) F, while its concentration remains constant (i.e., $[A] = [A]_0$). Horizontal dashed, dotted, and short dashed lines are the limit values for $[P]$, $[Q]$, and $[I]$ as the time approaches infinity, respectively. Initial conditions: $[A]_0 = 5$ mol $\cdot$m$^{-3}$, $[E]_0 = 1$ mol $\cdot$m$^{-3}$, $k_a = 2 \times 10^{-2}$ m$^3 \cdot$(mol$\cdot$ s)$^{-1}$, $k_d = 1.5 \times 10^{-2}$ m$^3 \cdot$(mol $\cdot$s)$^{-1}$, and $k_i = 5 \times 10^{-3}$ m$^3 \cdot$(mol s)$^{-1}$.

For the case in which E is inactivated, the values for $[P]$, $[Q]$, and $[I]$ are as follows:

$$[P]_\infty = \lim_{t \to \infty}[P] = \frac{k_a}{k_i}[E]_0, \tag{97}$$

$$[Q]_\infty = \lim_{t \to \infty}[Q] = \frac{k_a}{k_i}[E]_0, \tag{98}$$

$$[I]_\infty = \lim_{t \to \infty}[I] = [E]_0. \tag{99}$$

The limits for $[P]$ and $[Q]$ depend on both $[E]_0$ and the ratio (or catalytic turnover) between the rate constants $k_a$ and $k_i$ of Reactions (67) and (69), in which the enzyme state inactivated by the suicide substrate (i.e., E) participates. Both limits give the same value, implying that $[P]$ and $[Q]$ might first evolve differently, however they tend to approach the same limit concentration when the enzyme reaction stops. Thus, the ratio between $[P]_\infty$ and $[Q]_\infty$ is the unit regardless of the values for $k_a$ and $k_i$.

On comparing the results of the above limits (Equations (97)–(99)) with those of $[P]$, $[Q]$, and $[I]$ (Equations (100)–(102)) when the inactivated state is F, it can be newly observed that the limits similarly depend on both $[E]_0$ and the ratio between the rate constants $k_d$ and $k_i$ of Reactions (77) and (78), in which F participates. However, the values for the limits of $[P]_\infty$ and $[Q]_\infty$ are not the same.

$$[P]_\infty = \lim_{t \to \infty}[P] = \left(1 + \frac{k_d}{k_i}\right)[E]_0, \tag{100}$$

$$[Q]_\infty = \lim_{t \to \infty}[Q] = \frac{k_d}{k_i}[E]_0, \tag{101}$$

$$[I]_\infty = \lim_{t \to \infty}[I] = [E]_0. \tag{102}$$

The difference in the limits for $[P]_\infty$ and $[Q]_\infty$ implies that the ratio between $[P]_\infty$ and $[Q]_\infty$ thus, depends on $k_d$ and $k_i$, and it does not approach the unit unless $k_d >> k_i$.

Therefore, the analytical solutions for [P] and [Q] as time approaches infinity can be used to identify the enzyme state that becomes inactivated by the suicide substrate in an enzyme-catalyzed ping-pong reaction in which the suicide substrate undergoes disproportionation during the catalytic turnover. This kinetic analysis differs from the study presented by Varon et al. [32], in which approximate analytical solutions for the two-step suicide substrate inactivation of an enzyme-catalyzed ping-pong reaction with two substrates A and B were given. In the former study, the suicide substrate (A or B) was first defined, and hence, the enzyme state undergoing inactivation. In addition, the formation and accumulation of the substrate–enzyme complexes [EA] and [FB] were considered in the kinetic analysis as part of the two-step mechanism in which the (suicide) substrates first reversibly bound to the enzyme's active states. The former authors imposed the condition that the initial concentrations of the two substrates were much higher than that of the enzyme in order to linearize the ODE systems and analytically solve them. In the study here presented, the condition that the accumulation of [EA] and [FA] was negligible also allowed for solving the ODE system regardless of whether $[A]_0 >> [E]_0$ or $[A]_0 \approx [E]_0$. Therefore, despite the evident dissimilarities between both kinetic schemes and the conditions imposed to solve the ODE systems of the one-step and two-step suicide substrate inactivation mechanisms, the two studies reached equivalent conclusions about the enzyme state that became inactivated as time approached infinity, revealing the robustness of this type of exploratory analysis, even when some of the analytical solutions for the participating compounds are unwieldy.

## 4. Conclusions

The theoretical kinetic analysis here presented of the one-step suicide substrate inactivation of an enzyme-catalyzed ping-pong reaction with one substrate undergoing disproportionation provides approximate analytically integrated solutions for the time-dependent variation of the concentration for the participating compounds. To analytically solve the ODE system of the reactions, the reactant stationary approximation was invoked to ensure that the substrate–enzyme complexes did not steadily accumulate, and thus, their concentrations were negligible. These kinetic conditions were considered to occur experimentally for some enzymes such as catalase. Despite the unwieldiness of the approximate analytically integrated solutions, they were operational to qualitatively explore which enzyme state became inactivated by the suicide substrate. This type of kinetic analysis can be of potential use for the design of new drugs targeting enzyme-catalyzed ping-pong reactions. Additionally, the step-by-step theoretical approach followed in this study also has the purpose of showing—to those who are not conversant with the kinetic analysis of enzyme-catalyzed reactions—how both the standard quasi-steady-state and reactant stationary approximations are applied to analytically solve the ODE systems of enzyme reactions following the Michaelis–Menten mechanism.

**Supplementary Materials:** The following supporting information can be downloaded at https://www.mdpi.com/article/10.3390/math10224240/s1: Figure S1. Representative numerical (dashed lines, subscript *ns*) and analytical (solid lines, subscript *ss*) solutions of the time-dependent variation of (**A–D**) the concentration for the substrate A, the products P and Q, and the two active enzyme states E and F, and (**E,F**) the reaction rate for A, P, and Q of an enzyme-catalyzed ping-pong reaction; Figure S2. Representative numerically (subscript *ns*) and analytically (subscript $n = 1$ or 2) integrated solutions of the time-dependent variation of the concentration for (**A**) the substrate A and (**B**) the two active enzyme states E and F of an enzyme catalyzed ping-pong reaction under non-steady-state conditions. Wolfram Mathematica scripts for the analysis of the numerical and analytical solutions of the linear and nonlinear ODE systems and the graphical representations of the enzyme reaction models are available as Wolfram notebooks (Figures S1A and S2A,B) and pdf files.

**Author Contributions:** R.M. and J.B.A. conceptualized and designed the theoretical study; I.G.-F., O.B., N.B.-R., E.L.M.-B. and R.M. contributed to the supervision and visualization of the mathematical analysis of the ODE systems and graph presentation; R.M. and J.B.A. were responsible for funding acquisition; J.B.A. wrote the paper. All authors have read and agreed to the published version of the manuscript.

**Funding:** This research was funded by MCIN/AEI/10.13039/501100011033, grant number PID2019-107154RB-100, and the regional government of Castilla y León, grant number CSI260P20. The Project "CLU-2019-05—IRNASA/CSIC Unit of Excellence", funded by the Junta de Castilla y León and co-financed by the European Union (ERDF "Europe drives our growth"), and the CSIC Interdisciplinary Thematic Platform (PTI) Optimization of Agricultural and Forestry Systems (PTI-AGROFOR) are also acknowledged.

**Data Availability Statement:** Not applicable.

**Acknowledgments:** I.G.-F. holds a contract funded by the project CLU-2019-05, O.B. holds an Algerian Government fellowship, N.B.-R. is the recipient of a predoctoral contract from the Junta de Castilla y León (reference number E-37-2020-0042432), and E.L.M.-B. holds a postdoctoral contract funded by the project PID2019-107154RB-100. The support of the publication fee by the CSIC Open Access Publication Support Initiative, through its Unit of Information Resources for Research (URICI), is also acknowledged.

**Conflicts of Interest:** The authors declare no conflict of interest.

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
