# Peer review of "One-Step Suicide Substrate Inactivation Kinetics of a Ping-Pong Reaction with One Substrate Undergoing Disproportionation: A Theoretical Approach with Approximate Solutions"

_mathematics, doi:10.3390/math10224240_

Round 1

Reviewer 1 Report

Please, find the attached pdf file

Reviewer 2 Report

Comments concerning manuscript Mathematics-1964681

General comments:

The authors present a work focused on the theoretical kinetic analysis of one-step suicide substrate inactivation used to identify the enzyme state that becomes inactivated in a catalyzed-enzyme ping-pong reaction with one substrate undergoing disproportionation. The subject is interesting, but the manuscript quality is poor and an important revision is essential. The manuscript is inappropriate for publication in this Journal under its present form.

To help the authors in the eventual revision of the manuscript, some relevant comments are listed below:

-      In some cases, the nomenclature that appears in the equations is explained before them and in others after them. To avoid this inconvenience, add Sections of Nomenclature and Abbreviations. Remove them from the text. Please, put the units of each variable.

-      Improve the English of the manuscript.

-      For equations with too many terms, use two lines so they extend beyond the page margins. Respect the format.

Specific comments:

Abstract

-      Clearly state the objective of the paper and its results. Highlight novelty.

Introduction

-      In this section, there is information that is part of materials and methods. In addition, results are mentioned. Rewrite this section. The introduction must detail the work carried out on the subject by other authors or by the authors of this work, and emphasize the objective of the paper considering its novelty. 

-      Add a logic diagram.

-      Do not write in the first person.

Materials and Methods

-      The authors said: The Wolfram Mathematica scripts for the analysis of the 154 numerical and analytical solutions of the linear and non-linear ODE systems and 155 graphical representations of the enzyme reaction models are available from the corresponding author upon request. This information should be available in the supplementary material.

Results and Discussion

-      Figure 1. Improve the caption, write:  variation of reaction rate vs t for a)....and b)

-      Figures should be placed immediately after being mentioned in the text.

-      In the caption of all the figures, there is too much detail that should be clarified in the text.

Conclusions

-      Rewrite the conclusions considering the contributions of this work

Final assessment: Major Revision is required.

Round 2

Reviewer 2 Report

The work is accepted in the present form.